# A Study on Differential Biomarkers in the Milk of Holstein Cows with Different Somatic Cells Count Levels

**DOI:** 10.3390/ani13152446

**Published:** 2023-07-28

**Authors:** Yuanhang She, Jianying Liu, Minqiang Su, Yaokun Li, Yongqing Guo, Guangbin Liu, Ming Deng, Hongxian Qin, Baoli Sun, Jianchao Guo, Dewu Liu

**Affiliations:** 1College of Animal Sciences, South China Agricultural University, Guangzhou 510642, China; syh15521170780@163.com (Y.S.); smq2054113415@163.com (M.S.);; 2Agro-Tech Extension Center of Guangdong Province, Guangzhou 510500, China; ljy_1457@163.com (J.L.);; 3Guangdong Provincial Animal Husbandry Technology Promotion Station, Guangzhou 510500, China; 4Collaborative Innovation Center for Healthy Sheep Breeding and Zoonoses Prevention and Control, Shihezi University, Shihezi 832000, China

**Keywords:** Holstein cows, mastitis, different SCC levels, biomarkers

## Abstract

**Simple Summary:**

Mastitis is one of the most common diseases in the dairy industry in the world, and its treatment cost is also the most expensive of all diseases in the dairy industry. The research on the rapid diagnosis of cow mastitis is of great significance to the development of the dairy cow industry all over the world. In this study, we use microbiology and metabolomics techniques to explore the differential microbiota and metabolites in cow milk at different somatic levels, to find new biomarkers for the diagnosis of cow mastitis and provide a new plan for the diagnosis, prevention, and treatment of mastitis in the future.

**Abstract:**

Dairy cow mastitis is one of the common diseases of dairy cows, which will not only endanger the health of dairy cows but also affect the quality of milk. Dairy cow mastitis is an inflammatory reaction caused by pathogenic microorganisms and physical and chemical factors in dairy cow mammary glands. The number of SCC in the milk of dairy cows with different degrees of mastitis will increase in varying degrees. The rapid diagnosis of dairy cow mastitis is of great significance for dairy cow health and farm economy. Based on the results of many studies on the relationship between mastitis and somatic cell count in dairy cows, microflora, and metabolites in the milk of Holstein cows with low somatic cell level (SCC less than 200,000), medium somatic cell level (SCC up to 200,000 but less than 500,000) and high somatic cell level (SCC up to 5000,00) were analyzed by microbiome and metabolic group techniques. The results showed that there were significant differences in milk microbiota and metabolites among the three groups (*p* < 0.05), and there was a significant correlation between microbiota and metabolites. Meanwhile, in this experiment, 75 differential metabolites were identified in the H group and L group, 40 differential metabolites were identified in the M group and L group, and six differential microorganisms with LDA scores more than four were found in the H group and L group. These differential metabolites and differential microorganisms may become new biomarkers for the diagnosis, prevention, and treatment of cow mastitis in the future.

## 1. Introduction

Milk is the most important source of dairy products for human beings. It is one of the ideal natural foods for human beings [1]. In addition, milk not only contains nutrients provided by the body’s immune system but also contains many ingredients that can activate the immune system and improve immunity [2]. However, a variety of diseases of dairy cows will have a significant impact on the quality of milk, and in serious cases, it will also endanger the health of dairy cows. Mastitis is one of the most common diseases in the field of dairy cows in the world. It can cause huge economic consequences. The annual economic losses caused by mastitis amount to billions of dollars due to reduced milk production, discarded milk, increased knockout rates of dairy cows, and increased treatment costs [3]. Furthermore, Mastitis can lead to abnormal functions of embryos, ovaries, and uterus in dairy cows, resulting in a decrease in pregnancy rate and an increase in the risk of miscarriage. It has a great impact on embryo transfer in dairy cows [4]. In addition, in practical production, the use of antibiotics has been greatly restricted. In the United States, using antibiotic drugs to treat mastitis can cause the cow to lose its organic certification status. This poses a significant challenge for dairy farms in treating mastitis. Therefore, early diagnosis and prevention of mastitis are crucial [5].

The prominent feature of mastitis in cows is the presence of varying degrees of inflammation in the mammary glands, accompanied by a series of pathological changes. Among these changes, there is a profound relationship between the somatic cell count (SCC) in milk and mastitis in cows [6,7,8]. Dairy cow mastitis is divided into subclinical mastitis (SM) and clinical mastitis (CM) [9]. When cow mastitis occurs, the SCC in milk will increase. Most studies have shown that when 200,000 cells/mL is used as the cutoff value for the diagnosis of cow mastitis, the diagnostic error can be reduced to a minimum [10,11,12]. When SCC > 200,000 cells/mL was used as the threshold of dairy cow mastitis, the sensitivity (Se) and specificity (Sp) of identifying the main pathogens of dairy cow mastitis were 83.4% and 58.9%, respectively [13,14]. In addition, a series of studies have suggested that cows with SCC more than 200,000 will develop subclinical mastitis (SM) which leads to a decrease in milk production [15,16,17,18]. There are some studies that suggest that SCC < 500,000 cells/mL cows may suffer from SM [15,19,20]. Meanwhile, the SCC diagnostic threshold of clinical mastitis is SCC > 500,000 cells/mL [21].

Exploring a more efficient and automatic method for the diagnosis of dairy cow mastitis was an area of intense research during the 1960s [22]. With the advancement of diagnostic methods for somatic cell count (SCC), as well as its ability to diagnose mastitis rapidly and efficiently, SCC has been incorporated into the Dairy Herd Improvement (DHI) program [23]. At present, the number of somatic cells in milk has been proven to be an important parameter for the diagnosis of mastitis, especially for the diagnosis of subclinical mastitis [24]. However, the SCC diagnostic method also has some shortcomings. In the actual production process, in addition to pathogen infection, factors such as season, milking frequency, milking time, parity, and lactation stage of cows will all have an impact on the somatic cell count (SCC) value in milk [25]. Moreover, there are certain limitations in the detection method of somatic cell count (SCC) in milk, making it challenging to fully elucidate the pathogenesis, disease state, changes in microorganisms and metabolites in milk, and other related factors. As a result, it may be difficult to meet the research needs of scientists and researchers in this field [26,27]. Nowadays, SCC has become a widely accepted tool for the diagnosis of dairy cow mastitis. However, with the development of mastitis pathogens and various research needs, the research on mastitis needs to be explored at a deeper level [28].

In the late 1800s, there were studies that cow mastitis was caused by bacteria [29]. After hundreds of years of research, *Streptococcus agalactiae* and *Staphylococcus aureus* is the most prevalent pathogen causing mastitis [28]. In addition, studies have shown that several pathogens, including *Staphylococcus aureus*, *Streptococcus agalactiae*, and *Escherichia coli*, are commonly isolated and cultured from milk samples of cows with clinical mastitis [30]. At the same time, conditional pathogens such as *Enterobacteriaceae*, *Streptococcus*, *Lactococcus*, and *Prototheca* are often considered to be the pathogens of mastitis in modern dairy herds [30,31]. Additionally, when mastitis occurs in dairy cows, it will lead to alterations in various metabolites within the milk [32]. Studies have shown that some important metabolites in milk, including lactic acid, uracil, butyric acid, isoleucine, etc., are significantly correlated with SCC [33]. Therefore, the exploration of microorganisms and metabolites in milk can serve as an indicator of the health status of dairy cow udders and help identify new diagnostic biomarkers for mastitis [20,28].

In recent years, with the continuous development of more and more new technologies, the search for biomarkers of disease has become a major research boom. Biomarkers are indicators to detect the biological process and pathological state of the body, which can reveal a variety of health and disease characteristics [34]. According to the use of biomarkers, they can be divided into Diagnostic biomarkers, Monitoring biomarkers, Predictive biomarkers, and other types [34]. Biomarkers play an important role in disease diagnosis, drug development, and disease prevention and treatment [35]. Fatty acids (NEFA), beta-hydroxybutyrate (BHB), calcium and other biomarkers are commonly used to monitor the health status of dairy cows [36,37]. With the continuous advancement of high-throughput sequencing platforms, the methods for studying the composition and structure of microbial communities are constantly being optimized. This has greatly improved the identification of species in low-abundance communities and research on the integrity of microbial communities [38,39]. Currently, most studies on mastitis in cows focus on genomics, transcriptomics, and proteomics [40,41,42]. Untargeted metabolomics is a commonly used metabolomics research method. By detecting the metabolites present in the sample and obtaining quantitative information, significant differences can be identified between different groups of metabolites. This helps to explain the relationship between the detected metabolites and biological processes or states [43]. With the continuous improvement of metabolomics technology, more and more studies have begun to apply metabolomics to the analysis of milk samples [20,44].

In this experiment, microbiology and metabolomics techniques were used to study the raw milk of dairy cows with different somatic cells, aiming to explore the differences and correlations between microorganisms and metabolites in the milk from different somatic cells. It is expected that this study will assist researchers in identifying the differences in microorganisms and metabolites between healthy cows and cows with different somatic milk. This analysis will enable the identification of biomarkers for cow mastitis, providing a foundation for researchers to further understand the overall metabolic mechanism of cow mastitis. Moreover, it will offer new ideas for the rapid diagnosis and effective prevention of mastitis, ultimately improving animal welfare. Improve the output and quality of milk and reduce the economic loss of the dairy industry. An in-depth study of important differential metabolites and microorganisms in different somatic levels of milk has important theoretical significance.

## 2. Materials and Methods

### 2.1. Experimental Design and Samples Collection

In this experiment, based on the DHI records of a dairy farm in Guangdong Province, China over the past three months, 180 dairy cows were selected for routine milking. Before sampling, we first clean the cow’s udder using warm water at 40 °C; for the initial cleansing. Then, the cows’ four udders were washed with a 0.1% potassium permanganate solution. During the sampling process, the first three streams of milk were discarded, and then samples were taken from each of the cows’ four teats. Each milk sample collected per teat was 10 milliliters, and three samples were collected from each cow and placed in 50-milliliter sterile centrifuge tubes. After collecting the milk samples from each cow, they were quickly placed in a foam box containing dry ice for freezing. Once all the samples were collected, they were transported to the laboratory and frozen at −80 degrees Celsius. One of the samples collected from each cow was then taken out for SCC (somatic cell count) determination. Based on the SCC data, the cows’ previous history of mastitis, and their parity, 30 cows were selected for further analysis. Furthermore, 30 cows were selected and divided into 3 groups: 10 cows with low SCC (L group) (SCC less than 200,000 cells/mL); 10 cows with medium SCC (M group) (SCC up to 200,000 but less than 500,000 cells/mL), and 10 cows with high SCC (H group) (SCC up to 5,000,00 cells/mL).

### 2.2. Milk Microorganism DNA Extraction, PCR Amplification, and Sequencing

For the extraction and PCR amplification of microbial genomic DNA, the genomic DNA of the sample was extracted by CTAB or SDS method, and then the purity and concentration of DNA were detected by agarose gel electrophoresis. Take the selected 30 aliquots of DNA samples and transfer them into centrifuge tubes, then dilute the samples to a concentration of 1 ng/μL with sterile water. Using diluted genomic DNA as a template, according to the selection of sequencing region, PCR was performed using specific primers with Barcode, Phusion^®^ High-Fidelity PCR Master Mix with GC Buffer of New England Biolabs, and high efficiency and high-fidelity enzyme to ensure the efficiency and accuracy of amplification. The PCR products were detected by 2% agarose gel electrophoresis; the qualified PCR products were purified by magnetic beads, and the same number of samples were mixed according to the concentration of PCR products by enzyme labeling quantification. After being fully mixed, 2% agarose gel electrophoresis was used to detect PCR products, and the gel recovery kit provided by the Qiagen company was used to recover the products. Finally, the library was constructed by using TruSeq^®^ DNA PCR-Free Sample Preparation Kit library construction kit, and the constructed library was quantified by Qubit and Q-PCR. After the library was qualified, NovaSeq6000 was used for computer sequencing.

### 2.3. Metabolomics Analysis

A total of 30 samples were stored at −80 °C and were thawed on ice and vortexed for 10 s. A 150 μL extract solution (ACN: Methanol = 1:4, *V*/*V*) containing internal standard was added to a 50 microliter sample. Then, the sample was vortex for 3 min and centrifuged at 13,400 g for 10 min (4 °C). A 150 μL aliquot of the supernatant was collected and placed at −20 °C for 30 min, and then centrifuged at 13,400 g for 3 min (4 °C). 120 μL aliquots of supernatant were transferred for LC-MS analysis. All samples were acquired by the LC-MS system following machine orders. The analytical conditions were as follows, UPLC: column, Waters ACQUITY UPLC HSS T3 C18 (1.8 µm, 2.1 mm*100 mm); column temperature, 40 °C; flow rate, 0.4 mL/min; injection volume, 2 μL; solvent system, water (0.1% formic acid): acetonitrile (0.1% formic acid); The column was eluted with 5% mobile phase B (0.1% formic acid in acetonitrile) at 0 min followed by a linear gradient to 90% mobile phase B (0.1% formic acid in acetonitrile) over 11minutes, held for 1 min, and then come back to 5% mobile phase B within 0.1 min, held for 1.9 min.

### 2.4. Data Processing and Analyses

The sequencing data of the microbiome first split the sample data from the offline data according to the Barcode sequence and PCR amplification sequence, cut off the Barcode and primer sequence, and then use FLASH (v1.2.11) to assemble the reads of each sample. The splicing sequence obtained is the original Tags data (Raw Tags). The spliced Raw Tags needs to be strictly filtered to acquire high-quality Tags data (Clean Tags). Then, the Mathur method was used to analyze the species annotation with SILVA138.1 ‘s SSU rRNA database (the threshold was set to 0.8:1). The taxonomic information was obtained, and the community composition of each sample was calculated at each classification level: phylum, class, order, family, genus, and species. The phylogenetic relationships of all OTU/ASV representative sequences were obtained by fast multi-sequence alignment using MAFFT (v7.490) software. Finally, the data of each sample is homogenized, and the least amount of data in the sample is taken as the standard for homogenization processing. The subsequent Alpha diversity analysis and Beta diversity analysis are based on the homogenized data. The differences between the Alpha and Beta diversity index and dilution curve, PCA, PCoA, and NMDS diagrams were drawn by R software (Version 4.1.2), and functional annotation correlation analysis was carried out by Tax4Fun2.

The original data file acquired by LC-MS was converted into mzML and formatted by Proteo Wizard software (v3.0). Peak extraction, peak alignment, and retention time correction were respectively performed by the XCMS program. The “SVR” method was used to correct the peak area. The peaks with a detection rate lower than 50% in each group of samples were discarded. After that, metabolic identification information was obtained by searching the laboratory’s self-built database, integrated public database, AI database, and met DNA. Unsupervised PCA (principal component analysis) was performed by statistics function prcomp within R software. After that, metabolic identification information was obtained by searching the laboratory’s self-built database, integrated public database, AI database, and met DNA. Unsupervised PCA (principal component analysis) was performed by statistics function prcomp within R software. For two-group analysis, differential metabolites were determined by VIP (VIP > 1) and *p*-value (*p*-value < 0.05, Student’s *t*-test). VIP values were extracted from OPLS-DA results, which also contain score plots and permutation plots, and were generated using the R package Metabo AnalystR. The data was log transform (log2) and mean centering before OPLS-DA. Identified metabolites were annotated using the KEGG Compound database (http://www.kegg.jp/kegg/compound/, accessed on 3 March 2023), and annotated metabolites were then mapped to the KEGG Pathway database (http://www.kegg.jp/kegg/pathway.html, accessed on 3 March 2023). Significantly enriched pathways are identified with a hypergeometric test’s *p*-value for a given list of metabolites.

## 3. Results

### 3.1. Routine Analysis of Milk Composition

Based on the DHI test results, we selected 30 cows for further experimentation. Then, 10 cows in group L, group M, and group H were selected for further experiments. The milk composition and somatic cell test results were as follows (Table 1). In the table, “Fat” refers to milk fat content, “Pro” refers to milk protein content, “Lact” refers to lactose content, “Urea” refers to urea nitrogen content in milk, and “SCC” refers to somatic cell count in milk. From the results, it can be observed that the SCC values for groups H, M, and L all meet the requirements specified in the methodology.

### 3.2. Quality Control Results of Microbiological Group

The above 30 milk samples were extracted by DNA and amplified by PCR. One of the milk samples in group H was excluded after unqualified detection. A total of 2,280,991 original sequences were obtained from the remaining 29 milk samples by 16SrRNA gene sequencing. After quality control and filtration optimization, a total of 1,800,926 effective sequences were obtained. The average length read is 415 bp, and the details are shown in Table 2. The result of the dilution curve shows that with the increase of sequencing depth, the curve tends to flatten gradually, which shows that the sequencing data of this experiment is reasonable and can fully reflect the flora of the 3 groups of samples (Figure 1A). The effective sequences of the three groups of samples were clustered. A total of 7355 OTUs were obtained in the three groups, 5011 OTUs were obtained in the L group, 4691 OTUs were obtained in the M group, and 4083 OTUs were obtained in the H group, among which 2273 OTUs were shared by the three groups (Figure 1B).

### 3.3. Effects of Different Somatic Cell Levels on the Diversity of Microbiota in Milk Samples

Alpha Diversity of 16SrRNA gene sequencing was used to detect the changes of microflora in milk samples at different somatic cell levels. Shannon and Simpson index can reflect the diversity of the microbial community, while Chao1 and ACE index can reflect the abundance of the microbial community. Shannon index and Simpson diversity index showed that compared with the M and L groups, the intestinal microbial diversity of the H group was higher, and there was no significant difference between M and L groups. The richness estimation factors Chao1 and ACE showed that the total number of species and the number of OTU in the community were the most in group L and the least in group H, and the number of OTU and the total number of species in the microbial community decreased with the increase of somatic cell number (Table 3).

### 3.4. Effects of Different Somatic Cell Levels on the Composition of Microflora in Milk Samples

According to the results of species annotation, the species with the highest abundance of each group in the rumen of postpartum dairy cows were selected to generate a species relative abundance histogram (Figure 2A). The results showed that at the genus level, the dominant microflora in the milk of dairy cows with low SCC levels was *Enterobacter*, and with the increase of SCC, the relative abundance of *Enterobacter* decreased gradually, which was significantly lower than that of the H group, while the relative abundance of *Methyloversatilis* increased. In addition, compared with L group, the relative abundance of *Pseudomonas* in H group and M group increased significantly, while the relative abundance of *Ralstonia* decreased significantly. The relative abundance of *Enterococcus* in the H group was significantly lower than that in the L group and M group. The relative abundance of *Sphingomonas* and *Staphylococcus* increased significantly in H and M groups.

Non-Metric Multi-Dimensional Scaling (NMDS) is a nonlinear model based on the Bray-Curtis distance. According to the species information contained in the sample, it is reflected on the two-dimensional plane in the form of points. Using NMDS analysis, according to the species information contained in the sample, is reflected in the multi-dimensional space in the form of points, while the degree of difference between different samples is reflected by the distance between points, which can reflect the differences between groups and within groups of samples. As shown in Figure 2B, the 3 groups clustered separately based on analysis of similarities (ANOSIM method) (H versus L:R = 0.1641, *p*-value = 0.049; M versus L:R = 0.0336, *p*-value = 0.222; H versus M:R = 0.0.2026, *p*-value = 0.325). In addition, in order to study the similarity between different samples, hierarchical cluster analysis using UPGMA showed that most of the samples in group L were clustered in their own group, while most of the samples in groups H and M were mixed together (Figure 2C), indicating that the microbial species in group L were quite different from those in group M, while those in group H and group M were similar.

Using LEFSE analysis to compare the microbial communities in the milk of groups H and L, the results showed that a total of 6 differential microorganisms with an LDA score greater than 4 were identified between the H and L groups as biomarkers, including *Ralstonia* and *Bifidobacterium_adolescentis* in the L group, and *Pseudomonas_aeruginosa*, *Aeromonas_veronii*, *Acinetobacter_johnsonii* and Staphylococcus in group H. However, there were no biomarkers between group H and group M, group L and group M (Figure 3A). In addition, to find the different species between groups, a *t*-test test was carried out to find out the species with significant differences (*p* < 0.05). It was found that there was also a significant difference in *Bifidobacterium*, *Enterococcus*, and *Enterobacter* abundance between H and L groups (Figure 3B). Meanwhile, the abundance of *Jeotgalibaca*, *Aerococcus*, *Atopostipes*, *Facklamia*, *Erysipelothrix*, *Ruminobacter*, *Flavonifractor*, and *Paeniclostridium* in group M was significantly higher than that in group L. The abundance of *Tyzzerella*, *Dorea*, *Colidextribacter*, *Akkermansia*, *Lachnospira*, *Ruminococcus*, and unidentified *Lachnospiraceae* were significantly lower than that of the L group (Figure 4).

### 3.5. Functional Prediction of the Predominant Taxa

FAPROTAX function prediction linked microbial composition with function and found the metabolic pathways with significant differences (*p* < 0.05) through the *t*-test between groups. The results showed that a total of 19 grade 3 KEGG pathways were found to be differentially enriched between groups H and L. The abundance of nitrogen_respiration, nitrate_respiration, nitrite_respiration, aromatic_compound degradation, nitrous_oxide_denitrification, nitrite_denitrification, denitrification, nitrate_denitrification, dark_oxidation_of_sulfur_compounds, dark_thiosulfate_oxidation and nitrite_ammonification pathways of milk microbiome in group L was significantly lower than that in group H (Figure 5A). However, the abundance of human_pathogens_all, human_pathogens_meningitis, nitrate_reduction, human_associated, animal_parasites_or_symbionts, mammal_gut, and denitrification pathways was significantly higher than that of group H (*p* < 0.05) (Figure 5B).

### 3.6. Metabolome Results

The metabolomics characteristics of milk were determined by untargeted metabolomics. The PCA score plots showed no separation of metabolites in the H, M, and L groups (Figure 6). Therefore, the OPLS-DA model was used to further analyze the differences in metabolites among the groups. The OPLS-DA score showed that groups H, M, and L were divided into different regions (Figure 7A,C). Goodness-of-fit values and predictive ability values (H versus L group: R2X = 0.37, R2Y = 0.985, Q2 = 0.705, *p*-value < 0.05; M versus L group: R2X = 0.335, R2Y = 0.989, Q2 = 0.686, *p*-value < 0.05) indicated that the OPLS-DA model possessed a satisfactory fit with effective predictive power (Figure 7B,D).

Based on the results of OPLS-DA, our study used VIP as a threshold to further screen for differential metabolites between groups. In this study, metabolites with VIP ≥ 1.0 and *p*-value < 0.05 were defined as differential metabolites. Seventy-five different metabolites were detected in group H and group L. The levels of 39 metabolites increased significantly, while those of 36 metabolites decreased significantly, in the H group compared to the L group (Figure 8A). After qualitative and quantitative analysis of the detected metabolites, the differences in the quantitative information of metabolites in each group were compared in combination with the grouping of specific samples, and the top 20 metabolites in the comparison of different groups were drawn as a bar chart of differential metabolites. The levels of metabolites 2-Aminoethyl dihydrogen phosphate, 4,5-Dihydroxyphthalic acid, O-Succinyl-L-Homoserine and Isolithocholic acid in group L were significantly higher than those in group H. However, the levels of Tyrosine-Isoleucine, Dihydroxyacetone, D-fructofuranose, Dichlorprop, and Fludrocortisone acetate were significantly lower than those of group H (Figure 8B). Meanwhile, 40 different metabolites were detected in group M and group L. The levels of 16 metabolites increased significantly, while those of 24 metabolites decreased significantly, in the H group compared to the L group (Figure 9).

To determine the relevant metabolic pathways, enrichment of the KEGG pathway was performed according to the results of differential metabolites, where Rich Factor is the ratio of the number of differential metabolites in the corresponding pathway to the total number of metabolites annotated by the pathway. A larger value indicates a greater degree of enrichment. The results showed that the different metabolites in group H and group L were involved in 20 metabolic pathways, including Fructose and mannose metabolism, Taste transduction, Pyrimidine metabolism, 2-Oxocarboxylic acid metabolism, Nucleotide metabolism, Tryptophan metabolism, and ABC transporters. Differential metabolites of group M and group L are involved in Glycerolipid metabolism, Bile secretion, 2-Oxocarboxylic acid metabolism, and Carbon metabolism (Figure 10). The results showed that many pathway disorders were related to the progression of cow mastitis.

### 3.7. Correlation between Milk Microbiota and Metabolites at Different Somatic Cell Levels

Based on the Spearman correlation coefficient, we have found that the criteria for the significant correlation between differential microorganisms and metabolites were correlation |r| ≥ 0.7 and *p*-value < 0.05 for the significance test of the correlation coefficient. Spearman correlation coefficient and correlation test results of different microorganisms and metabolites at the gate level between group H and group L are as follows (Table 4). The results showed that the differential microbe *Actinobacteriota* and *Verrucomicrobiota* were significantly negatively correlated with the differential metabolite Cer(d18:0/23:0), benzoyl-coenzyme A, 1-*O*-Hexadecyl-sn-glycero-3-phosphocholine, and significantly positively correlated with the differential metabolite Thr-Trp-Met. In addition, there was a significant negative correlation between *Actinobacteriota* and Pro-Ala-Leu, Glu-His-His. It was also significantly positively associated with MG (0:0/18:2(9Z,12Z)/0:0), Dodecanedioic Aicd, MG(18:2/0:0/0:0), Glyceryl arachidonate and Hexadecanedioic acid. In addition, *Verrucomicrobiae* is also associated with Phenylacetylglycine, Lidocaine, prostaglandin E2 1-glyceryl ester, Ketoleucine, Prostaglandin h2, Isobutyryl CoA, and other differential metabolites showed a significant positive correlation. Meanwhile, there was a significant negative correlation between different microbiome *Deferribacteres* and different metabolites Pro-Gly-Val, NG-Amino-L-arginine, 5-Hydroxyisourate, 7-Hydroxywarfarin in group M and group L(Table 5).

## 4. Discussion

Previous studies have shown that the causes of mastitis infection in dairy cows are very complicated, and there are more than 150 pathogens, including bacteria, fungi, mycoplasma, and viruses. The changes in the structure and function of microflora in cow milk are considered to be related to the pathogenesis of many diseases. Due to the limitations of traditional microbial identification methods, about 25% of bacteria cannot be detected by routine culture in dairy cow mastitis cases [28,45]. Consistent with most reports, Consistent with most reports, the changes of microorganisms in milk during mastitis are characterized by the decrease of beneficial bacteria and the increase of harmful bacteria, which often lead to a decrease in microbial abundance and diversity [21,46]. This further illustrates the relationship between SCC and dairy cow mastitis. SCC is an important index for the diagnosis of dairy cow mastitis [17,47,48]. This may be because the pathogenic bacteria invaded the breast tissue, resulting in a sharp increase in the number of SCC, destroying the original microbial balance, making a small number of bacteria or some foreign strains in milk grow and multiply in large numbers, producing toxic metabolites, which led to the dominant position of the original symbiotic bacteria in milk [49,50]. Some studies have shown that the dominant bacteria that play an important role in the milk of healthy dairy cows and dairy cows with mastitis have no significant change at the gate level but have changed significantly at the genus level [51]. Wang et al. found that *Staphylococci* and *Streptococci* were the dominant bacteria in cow milk with clinical mastitis [21]. In this study, the relative abundance of *Staphylococcus* and *Sphingomonas* in the H group and M group was significantly higher than that in the L group. As a major foodborne pathogen, *Staphylococcus* is also one of the main pathogens of dairy cow mastitis. Studies have shown that in China and other countries, 10–40% of mastitis cases are caused by *Staphylococcus* [50]. *Enterobacter* belongs to general non-pathogenic or conditional pathogenic bacteria, which may infect the body and lead to inflammation only when the body’s immunity is low [52]. Interestingly, in this study, we found that *Enterobacter* was the dominant bacteria in group L, rather than *Lactobacillus* and other beneficial bacteria as most studies thought [53]. In addition, the relative abundance of *Bifidobacterium* in group L was significantly higher than that in groups H and M. *Bifidobacterium* is an important beneficial microorganism, that can inhibit the growth of harmful bacteria, resist the infection of pathogenic bacteria, and improve the disease resistance of animals. The decrease in its relative abundance will further increase the incidence of mastitis in dairy cows [54,55].

In addition, LEfSe analysis showed that the relative abundance of *Ralstonia* in group L was significantly higher than that in group H in the biomarkers of difference between group H and group L. Ralstonia is a widespread plant pathogen. So far, no research has shown that it is closely related to the occurrence of mastitis, which needs further study [56]. LEfSe analysis also showed that the relative abundance of *Pseudomonas_aeruginosa*, *Aeromonas_veronii*, and *Acinetobacter_johnsonii* in the H group was significantly higher than that in the L group. *Acinetobacter_johnsonii* is a gram-negative bacterium widely distributed in an aquatic environment, soil, and plants. It is mostly conditional pathogenic bacteria. It is easy to survive and reproduce in a moist environment, which leads to cow mastitis [57,58]. Ma et al. showed that there was a significant negative correlation between *Acinetobacter_johnsonii* and *Lactococcus*, and *Lactococcus* could inhibit the adhesion of common bovine mastitis pathogens to mammary gland cells and reduce the histological changes of mammary glands, so the antagonism of *Acinetobacter_johnsonii* and *Lactococcus* may indirectly aggravate the occurrence of mastitis [59,60]. *Pseudomonas_aeruginosa* is one of the pathogens of bovine mastitis. It is a kind of bacteria with low virulence, but it has the natural resistance to antibiotics and the ability to form biofilm, so it is difficult to treat this pathogen [61]. *Aeromonas_veronii* is a gram-negative bacterium widely distributed in aquatic environments, and it is a common conditional pathogen of fish-human livestock. Some species can cause diseases in humans, fish, and other aquatic animals, which can cause disease, inflammation, and infection in animals. However, few studies have shown that it is related to the occurrence of cow mastitis. In addition, *Aeromonas_veronii* can reduce nitrate to nitrite and change the metabolites in milk [62,63]. In addition, the relative abundance of microorganisms in the milk of group M was between group H and group L. Meanwhile, according to *t*-test results, the relative abundance of *Lachnospira*, *Akkermansia*, and *Ruminococcus* in group L was significantly higher than that in group M. These bacteria are recognized as beneficial bacteria. *Lachnospira* is a potentially beneficial bacteria that participates in the metabolism of a variety of carbohydrates and can increase the production of acetic acid and butyric acid, as well as water sister starch and other sugars to produce butyrate and short-chain fatty acids [64,65]. *Akkermansia* is considered a candidate for the next generation of important beneficial bacteria, which can improve host metabolic function and enhance host immune function [66,67]. Interestingly, the relative abundance of *Colidextribacter* in group L was significantly higher than that in group M. Colidextribacter is a common pathogen of mastitis [68]. Low SCC shows that the susceptibility to mastitis may be related to the characteristics of immune regulation. Liu et al. found that 1 has a significant correlation with subclinical mastitis in dairy cows [69]. This study also found that 1 the relative abundance in group M was significantly higher than that in group L. *Paeniclostridium*, a highly pathogenic bacterium found in medium-level SCC milk, can also cause mastitis [64]. However, the results of LEfSe analysis showed that the microorganisms not found in group M and group L could be used as biomarkers.

The metabolites in milk mainly come from the enzyme reaction in milk, the secretion of mammary epithelial cells, and the metabolism of different microorganisms in milk. Metabolic group data can provide further evidence for the changes in milk at different SCC levels [20]. In this study, we found that the contents of Tyrosine-Isoleucine, Dihydroxyacetonein, and 2,4-Dinitrotoluene in group H were significantly higher than those in group L. Grispoldi et al. found that there was a strong correlation between the content of Isoleucine and the relative abundance of *Staphylococcus aureus*. Similar to the findings of Grispoldi et al. [70]. Dihydroxyacetone, DHA is a kind of trisaccharide. Some studies have shown that metabolic imbalance and cellular stress are the result of DHA exposure. At the same time, several studies have proved the cytotoxicity and genotoxicity of DHA. In addition, once included in the metabolic pathway, DHA will induce the formation of AGE, thereby destroying proteins and lipids [71]. The 2,4-Dinitrotoluene is a common environmental pollutant, and its toxicity to mammals has been fully studied [72]. In addition, many studies have shown that changes in microbiota can affect the composition of whole-body metabolites [8,21,26,29]. In this study, there was also a significant correlation between microbiome and metabolite group in milk with different SCC levels. There was a significant correlation between *Actinobacteriota* and *Verrucomicrobiota* in group H and group L and differential metabolites such as phosphocholine and Thr-Trp-Met in benzoyl-coenzyme. *Verrucomicrobiota* is a newly identified bacteria, including a few identified species, mainly found in aquatic and soil environments or human faces. Some studies have shown that in colder environments, the proportion of rumen *Verrucomicrobiota* in cattle is higher, which can help the host improve energy conversion efficiency [73]. In addition, the first member and the only representative *Akkermansia muciniphila* of *Verrucomicrobiota* is an intestinal symbiote colonized in the mucous layer and is a promising candidate for probiotics, which is of great value in improving host metabolic function and immune response [67]. The metabolite Thr-Trp-Met, which has a significant positive correlation with *Verrucomicrobiota*, is an important amino acid in the growth of dairy cows and an important substance in protein synthesis in milk [74]. Furthermore, the most important role of *Actinobacteriota* is to produce antibiotics. Of the more than 2000 antibiotics found in the world, about 56% are produced by actinomycetes [75]. *Glycerylarachidonate*, which has a significant positive correlation with *Actinobacteriota*, can increase the postpartum feed intake of most mammals and increase the metabolic rate of dairy cows [76]. The significant positive correlation between beneficial bacteria and beneficial metabolites in milk further indicates that the composition of microorganisms in milk has a direct effect on the composition of metabolites.

## 5. Conclusions

In summary, the results of this study showed that there were significant differences in microorganisms and metabolites in milk with different SCC levels. 75 differential metabolites were identified in H group and L group, and 40 differential metabolites were identified in M group and L group. Meanwhile, this study identified 6 distinct microorganisms in both the L group and H group, with LDA scores higher than 4. These microorganisms could potentially serve as diagnostic biomarkers for mastitis in dairy cows. However, further testing and validation are required to confirm their accuracy as biomarkers. These differential microorganisms and metabolites may be potential biomarkers for the diagnosis of mastitis in dairy cows, which can provide a new scheme for the diagnosis, prevention, and treatment of mastitis in the future. In addition, according to the differences between microorganisms and metabolites in different SCC levels, the Spearman correlation coefficient was calculated, and it was found that there was a significant correlation between some microorganisms and metabolites in milk. In future research, the above differential microorganisms and metabolites can be further studied to create a new method for rapid diagnosis of dairy cow mastitis.

## Figures and Tables

**Figure 1 animals-13-02446-f001:**
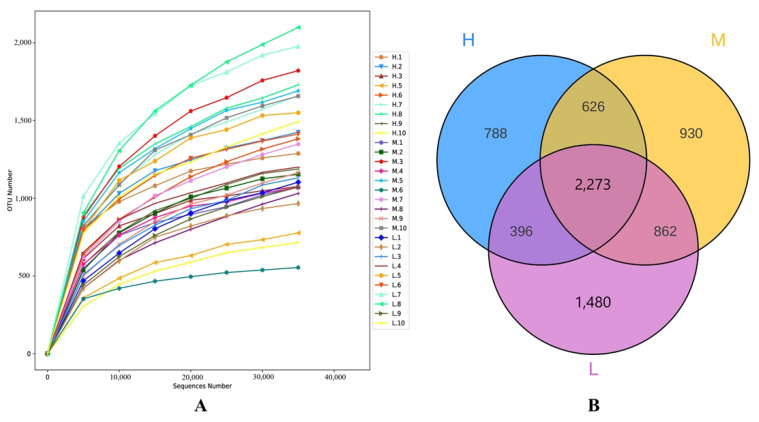
In the dilution curve, the Abscissa is the number of sequencing bars randomly selected from a sample, and the ordinate is the number of OTU that can be constructed based on the number of sequencing bars (**A**); each circle in the Wayne diagram represents a group of samples, the number of circles and the number of overlapping circles represents the common number of OTU between groups, and the number without overlapping represents the unique number of OTU of the group (**B**).

**Figure 2 animals-13-02446-f002:**
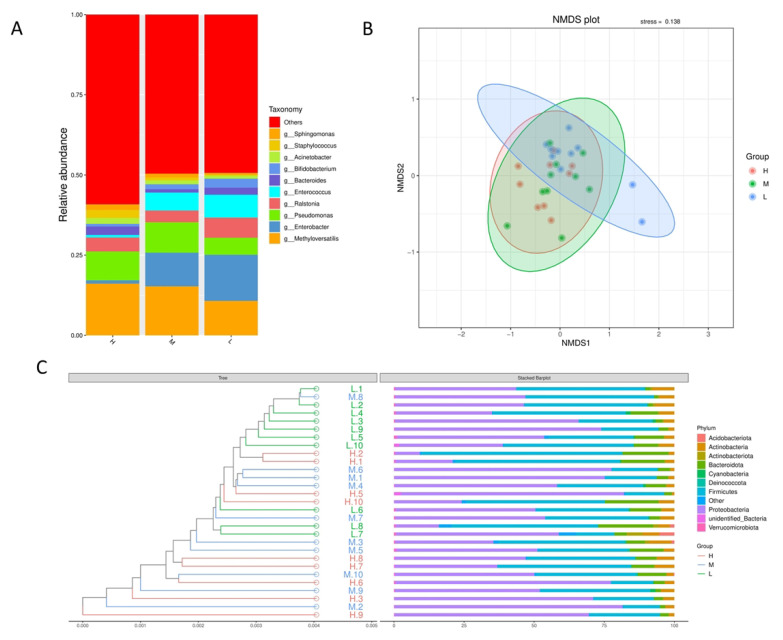
Composition of milk microbiome at different somatic cell levels. (H: *n* = 9, M: *n* = 10, L: *n* = 10). Composition and relative abundance of three groups of bacteria (**A**). β diversity (**B**) was measured by NMDS analysis. Each sample (**C**) is composed at the gate level using hierarchical clustering of unweighted Unifrac distances based on the OTU distribution.

**Figure 3 animals-13-02446-f003:**
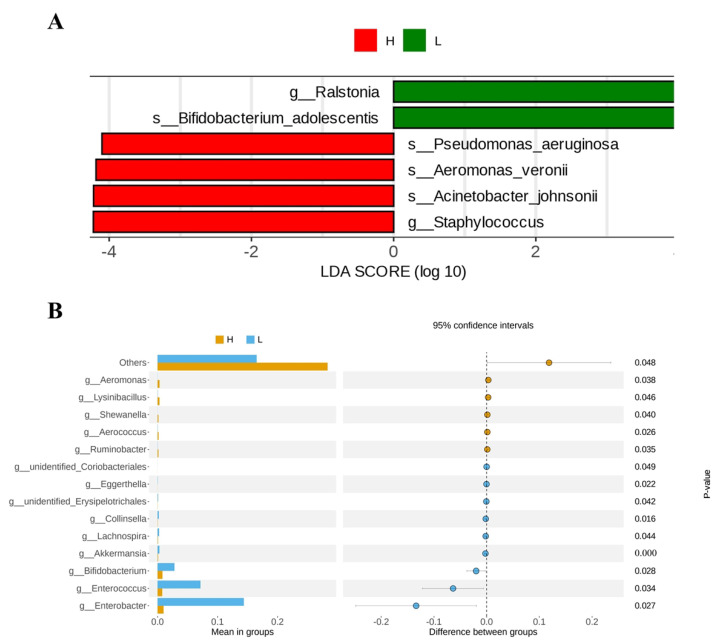
Dynamic changes of milk microbial communities at different somatic cell levels. By 16SrRNA sequencing (H: *n* = 9, L: *n* = 10), the milk microflora of H and L groups were represented by clad maps. Indicates the rich taxa in the L (green) and H (red) groups. The brightness of each dot is related to its LDA effect size (**A**). A *t*-test was used for statistical analysis of intestinal microflora in groups H and L (**B**).

**Figure 4 animals-13-02446-f004:**
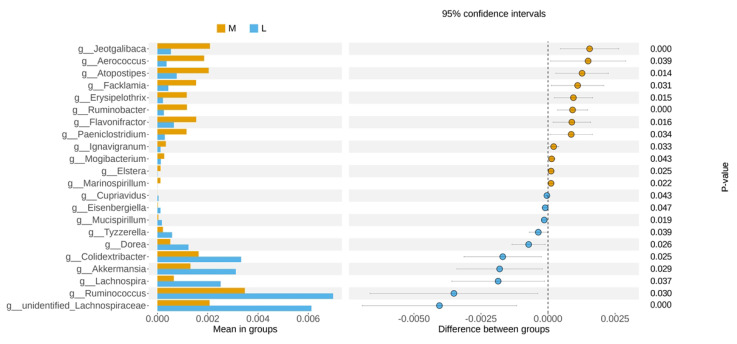
A *t*-test was used for statistical analysis of intestinal microflora in groups M and L.

**Figure 5 animals-13-02446-f005:**
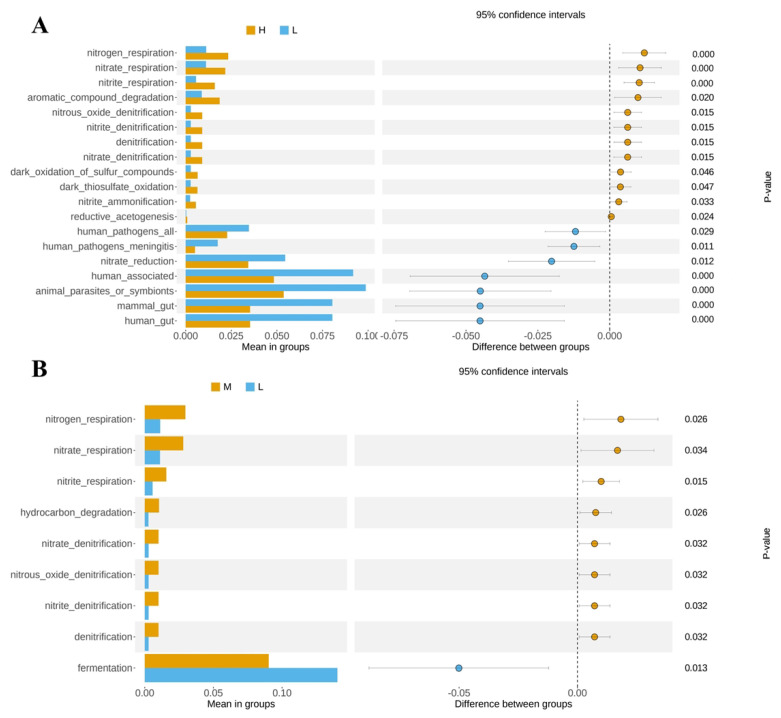
KEGG pathway in the milk of H and L groups (**A**). Microbial KEGG pathway in the milk of group M and group L (**B**).

**Figure 6 animals-13-02446-f006:**
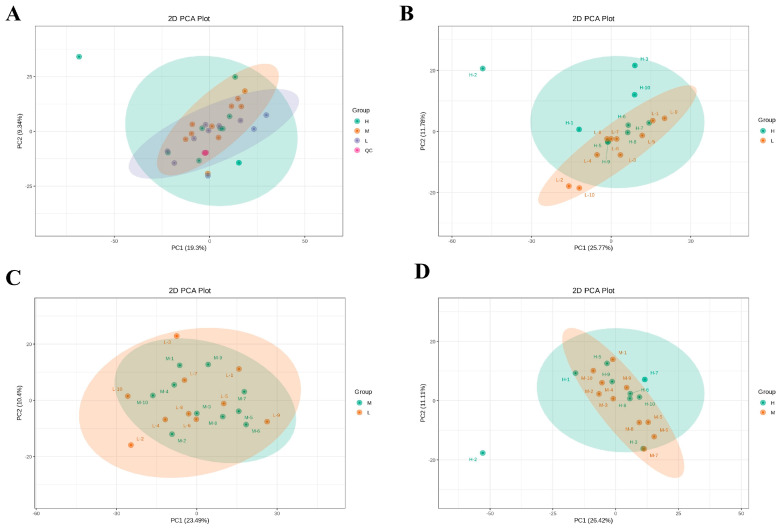
Each PCA map corresponds to a different grouping. PCA of milk metabolites structures in H, M, and L groups (**A**). PCA of milk metabolites structures in H and L groups (**B**). PCA of milk metabolites structures in M and L groups (**C**). PCA of milk metabolites structures in H and M groups (**D**). L: low SCC (*n* = 10), M: medium SCC (*n* = 10), and H: high SCC (*n* = 9).

**Figure 7 animals-13-02446-f007:**
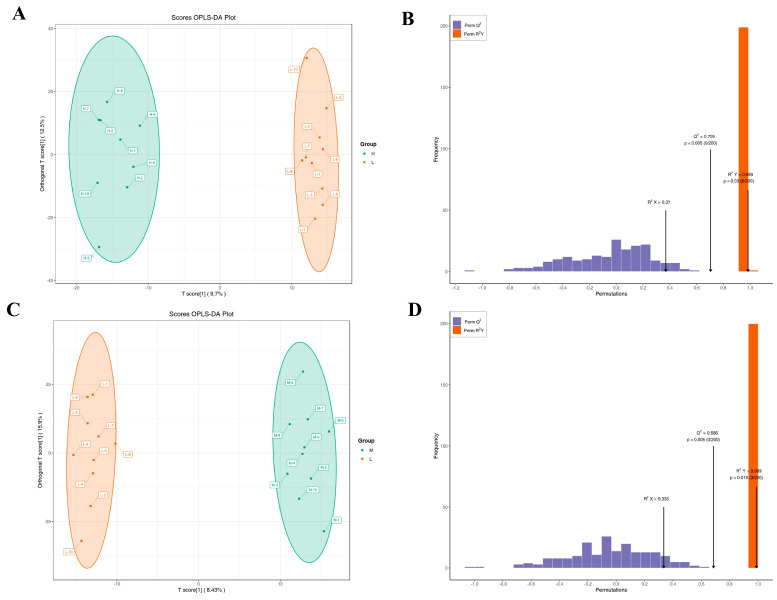
OPLS-DA analysis was performed to show the metabolite profile. OPLS-DA score plot and OPLS-DA model test chart showed good discrimination between H and L groups (**A**,**B**). OPLS-DA score plot and OPLS-DA model test chart showed good discrimination between M and L groups (**C**,**D**).

**Figure 8 animals-13-02446-f008:**
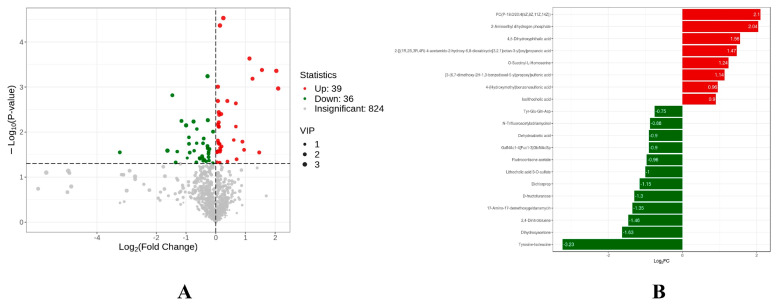
Volcanic polt of different metabolites between H and L groups (**A**). Linear discriminant analysis (LDA) effect size (LEfSe) results showed the top 20 differential metabolites between the H and L groups (**B**).

**Figure 9 animals-13-02446-f009:**
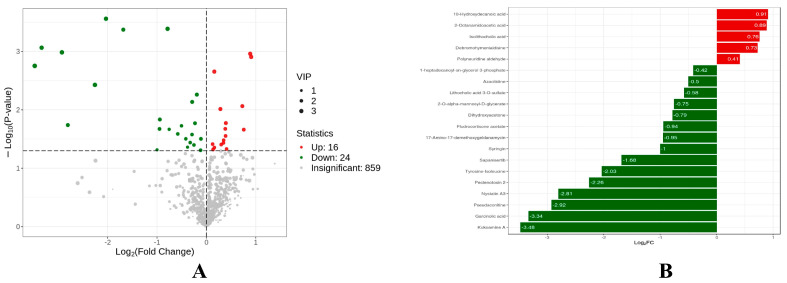
Volcanic polt of different metabolites between M and L groups (**A**). Linear discriminant analysis (LDA) effect size (LEfSe) results showed the top 20 differential metabolites between the M and L groups (**B**).

**Figure 10 animals-13-02446-f010:**
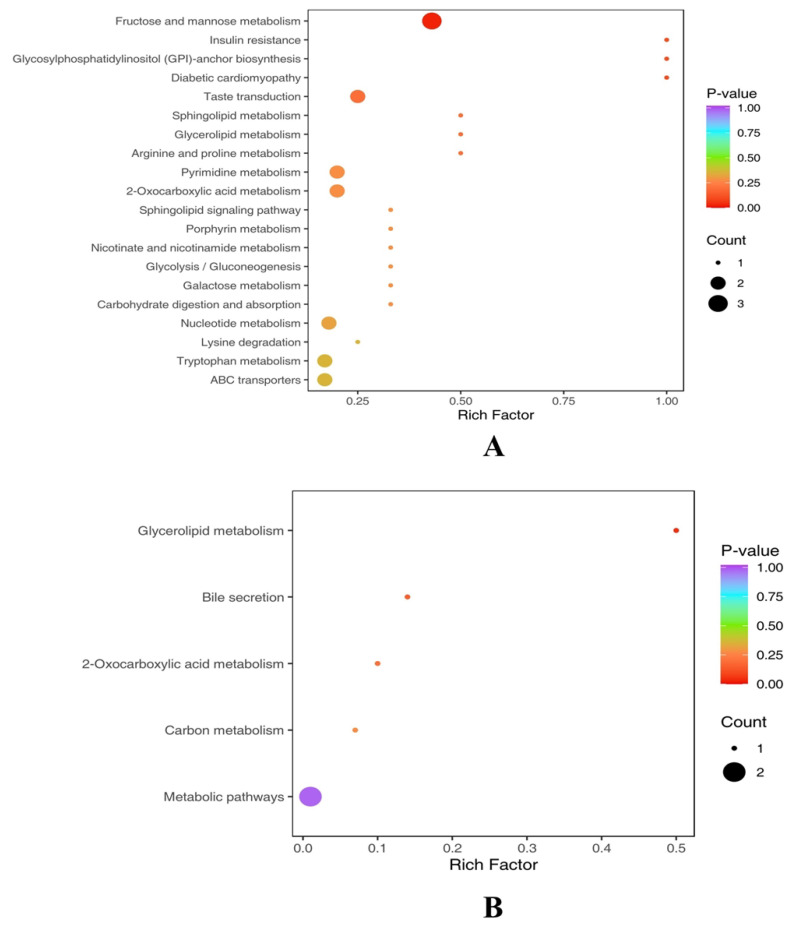
KEGG enrichment map of different metabolites between H and L groups (**A**). KEGG enrichment map of different metabolites between M and L groups (**B**).

**Table 1 animals-13-02446-t001:** Results of routine analysis of milk components.

Group	Fat (%)	Pro (%)	Lact (%)	Urea (%)	SCC (Cells/mL)
H	3.51	3.37	5.23	9.41	587,000
H	5.81	4.30	4.91	6.2	750,000
H	4.47	3.42	4.56	16.76	551,000
H	4.69	3.38	4.88	13.72	2,612,000
H	5.35	3.42	5.13	12.3	2,794,000
H	5.36	3.20	5.1	12.47	568,000
H	4.29	3.07	5.35	11.02	765,000
H	5.35	3.40	4.97	9.74	889,000
H	4.50	3.51	5.16	12.2	659,000
H	5.47	3.53	5.34	13.3	3,546,000
M	6.37	3.29	5.4	9.56	262,000
M	5.8	3.24	5.32	11.88	339,000
M	5.09	3.67	4.95	13.91	329,000
M	4.78	3.50	5.02	14.87	337,000
M	4.09	3.24	5.14	11.14	217,000
M	4.87	3.25	4.96	12.02	316,000
M	4.91	3.55	5.06	11.75	226,000
M	4.88	3.23	5.11	12.3	299,000
M	5.42	3.59	4.87	14.36	221,000
M	4.82	3.35	5.18	12.76	312,000
L	5.00	3.48	5.33	13.46	55,000
L	5.64	3.70	5.05	14.44	31,000
L	3.40	3.10	5.33	13.46	51,000
L	5.12	3.17	5.38	13.52	40,000
L	5.31	3.39	5.15	6.69	103,000
L	5.12	3.31	5.34	16.19	65,000
L	5.19	3.18	5	14.33	114,000
L	4.26	3.40	5.33	18.89	66,000
L	4.66	3.18	5.23	10.33	103,000
L	4.90	3.25	5.4	9.85	52,000

**Table 2 animals-13-02446-t002:** Illumina Nova Seq sequencing data.

Group	Raw Tags	Clean Tags	Effective Tags	Mean Length	GC (%)
H	763,556	634,829	568,851	409	53.21
M	786,328	700,067	616,384	417	53.61
L	731,107	712,575	615,691	420	54.10
Total	2,280,991	2,047,471	1,800,926	415	53.66

**Table 3 animals-13-02446-t003:** Alpha Diversity Statistics Table based on OTU.

Group	Shannon	Simpson	Chao1	ACE
H	6.53 ± 1.75	0.91 ± 0.09	1475.80 ± 383.74	1486.88 ± 380.89
M	5.97 ± 1.44	0.88 ± 0.08	1566.86 ± 492.09	1561.38 ± 499.96
L	6.10 ± 1.45	0.88 ± 0.09	1721.27 ± 432.20	1722.79 ± 441.51

**Table 4 animals-13-02446-t004:** Spearman correlation coefficient of different microorganisms and metabolites in H and L groups.

Index	Taxonomy	Correlation	*p*-Value	Compounds
MW0016991	p__Actinobacteriota	−0.7509	0.0002	Cer(d18:0/23:0)
MW0016991	p__Verrucomicrobiota	−0.7509	0.0002	Cer(d18:0/23:0)
MW0016070	p__Actinobacteriota	−0.7193	0.0005	benzoyl-coenzyme A
MW0016070	p__Verrucomicrobiota	−0.7070	0.0007	benzoyl-coenzyme A
MW0155767	p__Actinobacteriota	−0.7632	0.0001	Pro-Ala-Leu
MW0150025	p__Actinobacteriota	−0.7035	0.0008	Glu-His-His
MEDN0061	p__Verrucomicrobiota	0.8088	0.0000	Phenylacetylglycine
MW0007503	p__Verrucomicrobiota	0.8000	0.0000	Lidocaine
MW0011802	p__Verrucomicrobiota	0.7088	0.0007	prostaglandin E2 1-glyceryl ester
MW0157826	p__Actinobacteriota	0.7789	0.0001	Thr-Trp-Met
MW0157826	p__Verrucomicrobiota	0.7474	0.0002	Thr-Trp-Met
MW0105257	p__Verrucomicrobiota	0.7579	0.0002	Ketoleucine
MW0110625	p__Verrucomicrobiota	0.7123	0.0006	1-*O*-Hexadecyl-2-deoxy-2-thio-S-acetyl-sn-glyceryl-3-phosphorylcholine
MW0054970	p__Actinobacteriota	0.7684	0.0001	MG(0:0/18:2(9Z,12Z)/0:0)
MW0012825	p__Actinobacteriota	−0.7000	0.0008	1-*O*-Hexadecyl-sn-glycero-3-phosphocholine
MW0012825	p__Verrucomicrobiota	−0.7246	0.0005	1-*O*-Hexadecyl-sn-glycero-3-phosphocholine
MEDN0311	p__Actinobacteriota	0.7228	0.0005	Dodecanedioic Aicd
MW0055024	p__Actinobacteriota	0.7000	0.0008	MG(18:2/0:0/0:0)
MW0062272	p__Verrucomicrobiota	0.7509	0.0002	Prostaglandin h2
MW0012772	p__Actinobacteriota	0.7491	0.0002	Glyceryl arachidonate
MEDN0658	p__Actinobacteriota	0.7193	0.0005	Hexadecanedioic acid
MW0054060	p__Verrucomicrobiota	0.7018	0.0008	Isobutyryl CoA

**Table 5 animals-13-02446-t005:** Spearman correlation coefficient of different microorganisms and metabolites in M and L groups.

Index	Taxonomy	Correlation	*p*-Value	Compounds
MW0155873	p__Deferribacteres	−0.7262	0.0003	Pro-Gly-Val
MW0159163	Others	0.7910	0.0000	Val-Ser-Ser-Ser-Leu
MW0108785	p__Deferribacteres	−0.7490	0.0001	NG-Amino-l-arginine
MW0121539	p__Deferribacteres	−0.7323	0.0002	5-Hydroxyisourate
MW0112789	Others	0.7383	0.0002	3-Dehydrosphinganine
MW0056887	Others	−0.7880	0.0000	1-Palmitoyl-2-oleoyl-sn-glycero-3-phosphocholine
MW0137085	p__Deferribacteres	−0.7132	0.0004	7-Hydroxywarfarin

## Data Availability

Not applicable.

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
