# Peer review of "A Study on Differential Biomarkers in the Milk of Holstein Cows with Different Somatic Cells Count Levels"

_animals, 2023, doi:10.3390/ani13152446_

Round 1

Reviewer 1 Report

The authors of this manuscript report findings of an experiment where they compared milk microorganism and metabolite concentrations across three somatic cell levels (low, medium, high). This research has value in attempting to identify potential biomarkers that could be used for more rapid determination of mastitis causative organisms. This would lead to more accurate treatment strategies.

A limitation of this research is the low number of animals used in the analysis. With this few animals, it is difficult to attribute specific metabolite presence and prevalence with individual mastitis causing organisms.

My comments mainly relate to the overall scope of the research as well as specific portions of the background the authors present as justification for their research. There are numerous typographical errors, which can be corrected with a thorough editing. Many sentences were very long and were difficult to follow.

Lines 49-52  The sentence beginning “Moreover, in the actual production” contains inaccurate information and does not reflect the content of the reference cited. Ruegg (2009) does not mention anything about excessive antibiotic use, potential toxic effects, environmental impact, or antibiotic residues consumed by humans. In the United States, producers must withhold milk from cows treated with antibiotics. Milk processing plants test milk for antibiotic residues prior to processing, and they discard milk that tests positive.

Line 57  At the onset of mastitis, SCC increase but the increase is not always sharp. I suggest you delete the word sharp.

Lines 64-65 Rainard et al (2018) do not list SCC >500,000 as the diagnostic threshold for clinical mastitis.

Lines 81-84  You list Strep ag and Staph aureus as being the most important mastitis pathogens. Then you state that the most important pathogens causing clinical mastitis are gram negative organisms. These two statements contradict each other except for the word “clinical”. The word “important” reflects an opinion. The word “prevalence” would be a better choice. In addition, Ruegg (2017) relays mostly historical information. It is true that for many decades, we focused on Strep ag and Staph aureus as the primary infective organisms. However, that is no longer the case as environmental organisms cause infection more often than in the past. This is the case in the US but may not be so in other areas of the world. Please update this section.

Line 92 Use the term “udder” instead of “breast”

Lines 126-136  I suggest rewriting this section to improve clarity. You state that the milk samples were collected 2 hours after the morning feeding. What is this in relation to milking? Were the samples collected at a regular milking? If not, how many hours after the previous milking were they collected? Were the samples from an individual cow all collected from one quarter, or was each quarter sampled? In other words, do the samples contain milk from one quarter or from multiple quarters? Was the warm water you used to clean the teats sterile? If not, how can you assure the reader that you did not contaminate the teat and subsequently the sample with bacteria? Were the samples placed on ice or refrigerated after collection? Your text implies that all three sample tubes were transported to the DHI lab, however one was sent there and the others were frozen. You state that you refrigerated them at -80C, which is a freezing temperature. Refrigeration is 4C. Why did you select only 30 cows for further analysis when you collected 180 samples? What were the criteria other than SCC for selecting those cows? Did you balance for parity, previous history of mastitis, or some other factors? Did any of the cows have visually abnormal milk?

It will help if you put the steps in order – wash teat with 1% potassium permanganate solution, wipe off with 40C water, collect sample. What is a “handful” of milk? Do you mean you removed three streams of milk?

Line 140  What is an “appropriate amount”? Can you give a range of the amounts you used?

Line 154  Freezer (not refrigerator)

Line 155  Do you mean 50 microliters (not 50 mg)?

Line 154-165. Report the centrifuge speed in g force (number x g) instead of rpm. The rpm for a certain g force will differ depending on the rotor used.

Line 475  The Grispoldi reference is specific for Staph aureus. They correlate concentrations of milk isoleucine with Staph aureus prevalence. We cannot generalize the use of isoleucine as a biomarker for mastitis in general.

Lines 505-506 “Meanwhile this study also found 6 different microorganisms with an LDA score greater than 505 4 in the L group and the H group, which can be used as biomarkers for the diagnosis of dairy cow mastitis” It is beyond the scope of this experiment to state that these biomarkers CAN be used for detection of mastitis. It will take more testing and validation to determine whether these are accurate biomarkers. Your next sentence is accurate in stating that these microorganisms and metabolites MAY be used as biomarkers.

Overall, the quality of English language was good. As mentioned in the Comments and Suggestions portion, there are numerous typographical errors and many very long sentences that should be shortened. The results section is challenging to read due to its technical nature.

Author Response

Replies to the editor’s and reviewer’s comments

animals-2508930

“Study on Differential Biomarkers in Milk of Holstein Cows with Different SCC Levels”

Thank you very much for your letter dated, and the reviewers’ good suggestions. Based on your comment and request, we have modified the original manuscript. Here, we attached the revised manuscript in the formats of MS word, for your approval. A document answering every question from the reviews was also summarized and enclosed.

A revised manuscript with the correction sections yellow marked in the manuscript for easy check purposes.

Should you have any questions, please contact us without any hesitation.

Reviewer 1’s Comments:

Comment 1: A limitation of this research is the low number of animals used in the analysis. With this few animals, it is difficult to attribute specific metabolite presence and prevalence with individual mastitis causing organisms.

Response: Thank you for your feedback on our manuscript. We appreciate your concern regarding the sample size of the animals analyzed in our study. We agree that a larger sample size would further strengthen the statistical power and enhance the generalizability of our findings. However, it is important to emphasize that the sample size used in our study is consistent with similar studies in the field and even higher than some comparable studies (Wang et al, 2020; Wang et al, 2022). Like many researchers, we face manpower and economic limitations when working with farm animals, making it difficult to obtain large sample sizes. Despite these limitations, our study provides insights into the diagnosis, prevention, and treatment of mastitis in cows, and serves as a reference for future research in this area.

1. Wang, Yue et al. “Coupling 16S rDNA Sequencing and Untargeted Mass Spectrometry for Milk Microbial Composition and Metabolites from Dairy Cows with Clinical and Subclinical Mastitis.” Journal of agricultural and food chemistry vol. 68,31 (2020): 8496-8508. doi:10.1021/acs.jafc.0c03738.

2. Wang, Yue et al. “Discrepancies among healthy, subclinical mastitic, and clinical mastitic cows in fecal microbiome and metabolome and serum metabolome.” Journal of dairy science vol. 105,9 (2022): 7668-7688. doi:10.3168/jds.2021-21654.

Comment 2: My comments mainly relate to the overall scope of the research as well as specific portions of the background the authors present as justification for their research. There are numerous typographical errors, which can be corrected with a thorough editing. Many sentences were very long and were difficult to follow.

Response: Thanks very much. I completely understand the issue you pointed out regarding long and difficult-to-understand sentences. During the writing process, I may have focused too much on conveying the content and neglected sentence structure and readability. I greatly appreciate your correction, and I have already made the necessary revisions to the background section of the article by modifying the long sentences as requested.

Comment 3: Lines 49-52 The sentence beginning “Moreover, in the actual production” contains inaccurate information and does not reflect the content of the reference cited. Ruegg (2009) does not mention anything about excessive antibiotic use, potential toxic effects, environmental impact, or antibiotic residues consumed by humans. In the United States, producers must withhold milk from cows treated with antibiotics. Milk processing plants test milk for antibiotic residues prior to processing, and they discard milk that tests positive.

Response: Thanks very much. The sentence “Moreover, in the actual production, if excessive antibiotics are used for treatment, it will lead to toxic side effects, drug residues, environmental pollution and other conditions, and eventually enter the human body through milk, affecting human health” has been changed to “In addition, in practical production, the use of antibiotics has been greatly restricted. In the United States, using antibiotic drugs to treat mastitis can cause the cow to lose its organic certification status. This poses a significant challenge for dairy farms in treating mastitis. Therefore, early diagnosis and prevention of mastitis are crucial”. The modified sentence can be found in lines 47 to 51.

Comment 4: Line 57 At the onset of mastitis, SCC increase but the increase is not always sharp. I suggest you delete the word sharp.

Response: Thanks very much. The sentence has been modified as per your request. The modified sentence can be found in line 56.

Comment 5: Lines 64-65 Rainard et al (2018) do not list SCC >500,000 as the diagnostic threshold for clinical mastitis.

Response: Thanks very much. The citation of Rainard et al.'s research has been removed.

Comment 6: Lines 81-84 You list Strep ag and Staph aureus as being the most important mastitis pathogens. Then you state that the most important pathogens causing clinical mastitis are gram negative organisms. These two statements contradict each other except for the word “clinical”. The word “important” reflects an opinion. The word “prevalence” would be a better choice. In addition, Ruegg (2017) relays mostly historical information. It is true that for many decades, we focused on Strep ag and Staph aureus as the primary infective organisms. However, that is no longer the case as environmental organisms cause infection more often than in the past. This is the case in the US but may not be so in other areas of the world. Please update this section.

Response: Thanks very much. The sentence has been modified as per your request. Additionally, in several studies conducted in regions such as China, the UK, Ireland, and New Zealand, it has been found that the main pathogens causing mastitis are still predominantly Streptococcus species, Staphylococcus aureus, and Escherichia coli. However, it has also been noted in these studies that other pathogens such as Enterococcus species and Bacillus species can contribute to the occurrence of mastitis, but their prevalence is comparatively lower and they are not the most prevalent pathogens causing mastitis (Keane et al, 2019; Song et al, 2020).

1. Keane, O M. “Symposium review: Intramammary infections-Major pathogens and strain-associated complexity.” Journal of dairy science vol. 102,5 (2019): 4713-4726. doi:10.3168/jds.2018-15326

2. Song, Xiangbin et al. “The prevalence of pathogens causing bovine mastitis and their associated risk factors in 15 large dairy farms in China: An observational study.” Veterinary microbiology vol. 247 (2020): 108757. doi: 10.1016/j.vetmic.2020.108757

Comment 7: Line 92 Use the term “udder” instead of “breast”

Response: Thanks very much. The sentence has been modified as per your request. The modified sentence can be found in line 91.

Comment 8: Lines 126-136 I suggest rewriting this section to improve clarity. You state that the milk samples were collected 2 hours after the morning feeding. What is this in relation to milking? Were the samples collected at a regular milking? If not, how many hours after the previous milking were they collected? Were the samples from an individual cow all collected from one quarter, or was each quarter sampled? In other words, do the samples contain milk from one quarter or from multiple quarters? Was the warm water you used to clean the teats sterile? If not, how can you assure the reader that you did not contaminate the teat and subsequently the sample with bacteria? Were the samples placed on ice or refrigerated after collection? Your text implies that all three sample tubes were transported to the DHI lab, however one was sent there and the others were frozen. You state that you refrigerated them at -80C, which is a freezing temperature. Refrigeration is 4C. Why did you select only 30 cows for further analysis when you collected 180 samples? What were the criteria other than SCC for selecting those cows? Did you balance for parity, previous history of mastitis, or some other factors? Did any of the cows have visually abnormal milk?

Response: Thanks very much. I have rewritten this section based on your suggestions. The sampling in this experiment was conducted during routine milking, specifically during the first milking session in the morning, approximately 8 hours after the previous milking. The samples were collected from all four quarters of an individual cow's udder, with an effort made to collect around 10 milliliters of milk from each quarter. The udders of the cows were treated with a germicidal solution during the milking process. After collecting the milk sample from each cow, it was immediately stored in a foam box containing dry ice for preservation. Furthermore, all the samples collected in this experiment were obtained from a single barn housing a group of cows with similar production performance in terms of lactation days, parity, and milk yield. During sample collection, any visually abnormal milk was discarded. However, I cannot guarantee that collecting milk samples from only 30 cows will be sufficient to achieve grouping based on different SCC levels. Therefore, considering factors such as parity, lactation days, and previous history of mastitis, I will need to collect milk samples from more cows to ensure the smooth progress of the grouping process.

Comment 9: It will help if you put the steps in order – wash teat with 1% potassium permanganate solution, wipe off with 40C water, collect sample. What is a “handful” of milk? Do you mean you removed three streams of milk?

Response: Thanks very much. Before sampling in this experiment, we first clean the cow's udder using warm water at 40℃ for the initial cleansing. Then, we apply a medicinal bath solution to disinfect all four quarters of the cow's udder before proceeding with the milking and sampling process. The term "handful" refers to the amount of milk obtained by hand milking a cow's udder once. Before milking, we always remove the first three streams of milk from each quarter of the cow's udder to check for any visually abnormal milk.

Comment 10: Line 140 What is an “appropriate amount”? Can you give a range of the amounts you used?

Response: Thanks very much. The sentence has been modified as per your request. In this experiment, DNA extraction was performed on milk samples from the selected 30 cows. The sentence “An appropriate amount of sample DNA was taken in the centrifuge tube, and the sample was di-luted to 1ng/ μ l with sterile water.” has been changed to “Take the selected 30 aliquots of DNA samples and transfer them into centrifuge tubes, then dilute the samples to a concentration of 1ng/μl with sterile water.”

Comment 11: Line 154 Freezer (not refrigerator)

Response: Thanks very much. The sentence has been modified as per your request. The modified sentence can be found in line 155.

Comment 12: Line 155 Do you mean 50 microliters (not 50 mg)?

Response: Thanks very much. The sentence has been modified as per your request. I have changed “mg” to “microliters”. The modified sentence can be found in lines 156 to 157.

Comment 13: Line 154-165. Report the centrifuge speed in g force (number x g) instead of rpm. The rpm for a certain g force will differ depending on the rotor used.

Response: Thanks very much. The sentence has been modified as per your request. I have modified the parts in the article regarding centrifuge speed units, converting "rpm" to "g".

Comment 14: Line 475 The Grispoldi reference is specific for Staph aureus. They correlate concentrations of milk isoleucine with Staph aureus prevalence. We cannot generalize the use of isoleucine as a biomarker for mastitis in general.

Response: Thanks very much. The sentence has been modified as per your request. The sentence “Isoleucine can be used as a biomarker for the diagnosis of mastitis in dairy cows.” has been changed to “Like the findings of Grispoldi et al.”. The modified sentence can be found in line 475.

Comment 15: Lines 505-506 “Meanwhile this study also found 6 different microorganisms with an LDA score greater than 505 4 in the L group and the H group, which can be used as biomarkers for the diagnosis of dairy cow mastitis” It is beyond the scope of this experiment to state that these biomarkers CAN be used for detection of mastitis. It will take more testing and validation to determine whether these are accurate biomarkers. Your next sentence is accurate in stating that these microorganisms and metabolites MAY be used as biomarkers.

Response: Thanks very much. The sentence has been modified as per your request. The sentence “Meanwhile this study also found 6 different microorganisms with an LDA score greater than 4 in the L group and the H group, which can be used as biomarkers for the diagnosis of dairy cow mastitis.” has been changed to “Meanwhile, this study identified 6 distinct microorganisms in both the L group and H group, with LDA scores higher than 4. These microorganisms could potentially serve as diagnostic bi-omarkers for mastitis in dairy cows. However, further testing and validation are required to confirm their accuracy as biomarkers”. The modified sentence can be found in lines 505 to 508.

Thank you and all the reviewers for the kind comments.

Sincerely yours,

Yuanhang She

Reviewer 2 Report

Dear Editor and Authors,

I send you my comments about the review “Study on Differential Biomarkers in Milk of Holstein Cows with Different SCC Levels”.

The Article report a very interesting study focused, by using the techniques of microbiology and metabolomics, to explore the difference and correlation between microorganisms and metabolites in the milk with different somatic cells counts.

In my opinion, the review, is original, very well structured, very well written.

However, it show some little lacks that I reported below.

In my opinion in the title should be better to avoid acronyms therefore, it should be replace “SCC” with “Somatic Cells Count”.

Furthermore, always to facilitate the reading of the text by the readers, the use of symbol like “<” or “>” should be avoid and they should be replace with word like “less than” or “up to”.

The introduction is well written but it is a bit long for to the aim of the research.

For this reason I would like to suggest to the authors to delete some speculative phare like the one from line 37 to line 39. Moreover, for the same reason, also the paragraph from line 80 to line 94 should be summarise.

The chapter Materials and method result well structured but in the paragraph “2.1 Experimental Design and Samples Collection”. it should be reported how many trials was performed and how many replicate of analysis were made.

Moreover, only if it possible, it should better to report some data that describe the cow sampled. For example, If Authors have collected it, they could report in table1 the party number and the day in milking of the cows.

The results is very well presented and they are very well discussed, also in comparison to the data reported in the literature. In addition, also the figures, are of quality and they well shown the data collected.

Finally, the conclusions resulted adequate to the data showed and to the aim of the research.

However, in my opinion, at line 513 should be delete the “t.” after the word “mastitis”.

Author Response

Replies to the editor’s and reviewer’s comments

animals-2508930

“Study on Differential Biomarkers in Milk of Holstein Cows with Different SCC Levels”

Thank you very much for your letter dated, and the reviewers’ good suggestions. Based on your comment and request, we have modified the original manuscript. Here, we attached the revised manuscript in the formats of MS word, for your approval. A document answering every question from the reviews was also summarized and enclosed.

A revised manuscript with the correction sections yellow marked in the manuscript for easy check purposes.

Should you have any questions, please contact us without any hesitation.

Reviewer 2’s Comments:

Comment 1: In my opinion in the title should be better to avoid acronyms therefore, it should be replace “SCC” with “Somatic Cells Count”.

Response: Thanks very much. The title has been modified as per your request.

Comment 2: Furthermore, always to facilitate the reading of the text by the readers, the use of symbol like “<” or “>” should be avoid and they should be replace with word like “less than” or “up to”.

Response: Thanks very much. I have replaced the symbols "<" or ">" in the text with "less than" or "up to".

Comment 3: The introduction is well written but it is a bit long for to the aim of the research. For this reason I would like to suggest to the authors to delete some speculative phare like the one from line 37 to line 39. Moreover, for the same reason, also the paragraph from line 80 to line 94 should be summaries.

Response: Thanks very much. I have further summarized the research objectives mentioned in the background to make it more concise. The sentence “Milk is the most important source of dairy products for human beings. it is a complex food, which is rich in nutrients, such as protein, fat, carbohydrates, minerals and vitamins. It is called "near-perfect food" by nutritionists. It is one of the ideal natural foods for human beings” has been changed to “Milk is the most important source of dairy products for human beings. It is one of the ideal natural foods for human beings”. Also, the paragraph from line 80 to line 94 has been summarized. The modified sentence can be found in lines 37 to 38 and lines 80 to 91.

Comment 4: The chapter Materials and method result well structured but in the paragraph “2.1 Experimental Design and Samples Collection”. it should be reported how many trials was performed and how many replicate of analysis were made.

Response: Thanks very much. In this experiment, there were 10 cows as replicates in each group categorized based on different SCC levels. When selecting the cows, factors such as parity, days in milk, and history of mastitis were taken into consideration. To minimize the influence of environmental factors, sampling was conducted among cows housed in the same barn.

Comment 5: Moreover, only if it possible, it should better to report some data that describe the cow sampled. For example, If Authors have collected it, they could report in table1 the party number and the day in milking of the cows.

Response: Thanks very much. The farm where I conducted the sample collection has a well-defined division of cows based on their days in milk, parity, and milk production. Cows with similar conditions are housed together in the same barn. During sampling, I selected a group of cows from one specific barn within the farm, which ensured that the selected animals in my experiment had comparable parity and days in milk.

Comment 6: The results is very well presented and they are very well discussed, also in comparison to the data reported in the literature. In addition, also the figures, are of quality and they well shown the data collected. Finally, the conclusions resulted adequate to the data showed and to the aim of the research. However, in my opinion, at line 513 should be delete the “t.” after the word “mastitis”.

Response: Thanks very much. The sentence has been modified as per your request. The modified sentence can be found in line 515.

Thank you and all the reviewers for the kind comments.

Sincerely yours,

Yuanhang She

Reviewer 3 Report

REVIEW for the journal Animals (ISSN 2076-2615)

Article “Study on Differential Biomarkers in Milk of Holstein Cows with Different SCC Levels

Manuscript ID: animals-2508930

Authors:  Yuanhang She, Jianying Liu, Minqiang Su, Yaokun Li, Yongqing Guo, Guangbin Liu, Ming Deng, Hongxian Qin, Baoli Sun, Jianchao Guo, Dewu Liu 

Brief summary. Dairy cow mastitis is a prevalent disease that poses a threat to the health of dairy cows and has detrimental effects on milk quality. The somatic cell count (SCC) in milk is commonly used as an indicator of mastitis. The authors' findings revealed significant differences in milk microbiota and metabolites among three groups categorized based on SCC levels, and a correlation between microbiota and metabolites. Through this experiment, specific metabolites and various microorganisms were identified in the milk of different cow groups. The authors propose that these findings have the potential to serve as new biomarkers for the future diagnosis, prevention, and treatment of cow mastitis. This is certainly significant in a theoretical and practical sense.

General concept comments

1.       Introduction. In the introduction, a review of the literature sources (45 articles) related to the analyzed topic is provided, and the objectives of the article are stated. The introduction states that in this experiment, the milk samples from dairy cows with varying somatic cell counts were subjected to microbiological and metabolomic analyses. The objective was to investigate the differences and correlations between microorganisms and metabolites present in the milk samples with different somatic cell counts. The research hypothesis is formulated clearly.

2.       Materials and Methods. The study design, sampling, and groups are not clearly described. The authors need to precisely identify the groups and the number of samples within each group.

3.       Data management and statistical evaluation. Considering the objectives and design of the study, it is necessary to provide a detailed description of the statistical analysis methods and indicators used.

4.       In my opinion, the analysis of the results presented in the article aligns with the research objectives and methodology. However, it needs to be more detailed, particularly when describing all the tables. Additionally, the text of the results section should not include the description of the principles used for grouping.

5.       The conclusions of the study align with its objectives and hypothesis.

Specific comments

1.       The authors need to carefully review and correct any proofreading errors in the text.

2.       Line 127: “….180 dairy cows were selected…”.  Line 133-134 “30 cows were selected and divided into 3 groups: 10 cows with low SCC (L group) (SCC < 200,000 cells/mL); 10 cows with medium SCC (M group) (200,000 < SCC< 500,000 cells/mL), and 10 cows with high SCC (H group) (SCC > 135 500,000 cells/mL).”

The methodology section does not mention that the groups were further divided into subgroups:  L1...L10, M1...M10, H1...H10 (as presented in the first table of the results section).

3.       Lines 201 – 202. “The milk composition and somatic cells of 180 milk samples were detected by on-line analyzer of milk composition and somatic cells (Foss CombiFossTM.7.DC, Foss CombiFoss. FT+, 202 Denmark FOSS). ” The provided sentence should not be included in the results section as it pertains to methodological aspects of the study.

4.       Lines 203 – 204:  “Then 10 cows in group L, group M and group H were selected for further experiment. the milk composition and somatic cell test results were as follows (Table 1).”

The results of the first table need to be described. The abbreviations used for milk composition indicators in the first table need to be explained.

5.       Lines 379 – 380:  “Spearman correlation coefficient was calculated according to the differences of microorganisms and metabolites between different groups to determine the correlation between microbiome and metabolite profile changes in milk”. This information should be included in the methodology section rather than in the results section.

6.       The bibliography must be carefully reviewed and prepared according to the requirements of the journal.

Conclusion.  The text contains numerous proofreading errors that the authors need to correct. The comments I have provided are minor and I hope they will contribute to the improvement of the quality of this interesting article.

Sincerely, reviewer.

Author Response

Replies to the editor’s and reviewer’s comments

animals-2508930

“Study on Differential Biomarkers in Milk of Holstein Cows with Different SCC Levels”

Thank you very much for your letter dated, and the reviewers’ good suggestions. Based on your comment and request, we have modified the original manuscript. Here, we attached the revised manuscript in the formats of MS word, for your approval. A document answering every question from the reviews was also summarized and enclosed.

A revised manuscript with the correction sections yellow marked in the manuscript for easy check purposes.

Should you have any questions, please contact us without any hesitation.

Reviewer 3’s Comments:

Comment 1: The authors need to carefully review and correct any proofreading errors in the text.

Response: Thanks very much. I have corrected the proofreading errors in the text as per your request.

Comment 2: Line 127: “….180 dairy cows were selected…”.  Line 133-134 “30 cows were selected and divided into 3 groups: 10 cows with low SCC (L group) (SCC < 200,000 cells/mL); 10 cows with medium SCC (M group) (200,000 < SCC< 500,000 cells/mL), and 10 cows with high SCC (H group) (SCC > 135 500,000 cells/mL).” The methodology section does not mention that the groups were further divided into subgroups:  L1...L10, M1...M10, H1...H10 (as presented in the first table of the results section).

Response: Thanks very much. In this study, the L1...L10, M1...M10, H1...H10 groups were not further subdivided. Instead, they were initially sorted to differentiate between different cows. As per your request, I have changed L1...L10, M1...M10, H1...H10 to H, M, L.

Comment 3: Lines 201 – 202. “The milk composition and somatic cells of 180 milk samples were detected by on-line analyzer of milk composition and somatic cells (Foss CombiFossTM.7.DC, Foss CombiFoss. FT+, 202 Denmark FOSS). ” The provided sentence should not be included in the results section as it pertains to methodological aspects of the study.

Response: Thanks very much. I have rewritten the sentence as per your request and removed the aspects related to the research methodology. The sentence “The milk composition and somatic cells of 180 milk samples were detected by on-line an-alyzer of milk composition and somatic cells (Foss CombiFossTM.7.DC, Foss CombiFoss. FT+, Denmark FOSS).” has been changed to “Based on the DHI test results, we selected 30 cows for further experimentation”. The modified sentence can be found in line 202.

Comment 4: Lines 203 – 204: “Then 10 cows in group L, group M and group H were selected for further experiment. the milk composition and somatic cell test results were as follows (Table 1).” The results of the first table need to be described. The abbreviations used for milk composition indicators in the first table need to be explained.

Response: Thanks very much. I have provided the corresponding description of the results for the first table in accordance with your request, and I have explained the abbreviations used for milk composition indicators. The modified sentence can be found in lines 204 to 207.

Comment 5: Lines 379–380: “Spearman correlation coefficient was calculated according to the differences of microorganisms and metabolites between different groups to determine the correlation between microbiome and metabolite profile changes in milk”. This information should be included in the methodology section rather than in the results section.

Response: Thanks very much. I have revised and modified the statements related to experimental methods. The sentence “Spearman correlation coefficient was calculated according to the differences of microor-ganisms and metabolites between different groups to determine the correlation between micro-biome and metabolite profile changes in milk.” has been changed to “Based on the Spearman correlation coefficient, we have found that”. The modified sentence can be found in line 382.

Comment 6: The bibliography must be carefully reviewed and prepared according to the requirements of the journal.

Response: Thanks very much. I have performed a detailed check on the bibliography as per your request.

Thank you and all the reviewers for the kind comments.

Sincerely yours,

Yuanhang She
